# Comparison of Secular Trends in Esophageal Cancer Mortality in China and Japan during 1990–2019: An Age-Period-Cohort Analysis

**DOI:** 10.3390/ijerph191610302

**Published:** 2022-08-18

**Authors:** Ruiqing Li, Jinyi Sun, Tong Wang, Lihong Huang, Shuwen Wang, Panglin Sun, Chuanhua Yu

**Affiliations:** 1Department of Epidemiology and Biostatistics, School of Public Health, Wuhan University, Wuhan 430071, China; 2Global Health Institute, Wuhan University, Wuhan 430071, China

**Keywords:** esophageal cancer, mortality, age-period-cohort model

## Abstract

Esophageal cancer is a prevalent and often fatal malignancy all over the world, with China and Japan bearing a disproportionately high burden. Consequently, we explored and compared the long-term changes in esophageal cancer mortality in China and Japan from 1990 to 2019 to see if there were any etiological clues. From 1990 to 2019, data on mortality in China and Japan were gathered from the Global Burden of Disease Study 2019 (GBD 2019). The age-period-cohort (APC) model was utilized to evaluate the effects of age, period, and cohort. Between 1990 and 2019, the age-standardized mortality rates (ASMRs) for esophageal cancer fell in both nations, with China showing a tremendous reduction after 2005. The overall net drifts per year were more impressive in China (−5.22% [95% CI, −5.77 to −4.68] for females, −1.98% [−2.22 to −1.74] for males) than in Japan (−0.50% [−0.91 to −0.08] for females, −1.86% [−2.12 to −1.59] for males), and the local drift values in both countries were less than zero in all age groups for both sexes. The longitudinal age curves of esophageal cancer mortality increased as age advances and the sex disparity gradually exacerbates with age. The period and cohort effects were uncovered to have similar declining patterns for both sexes in both nations; however, the improvement of cohort effects for China’s younger generation has stagnated. The ASMRs, period effects, and cohort effects have decreased for both countries and sexes over the 1990–2019 period. The decline in cohort effects for China’s younger generation has plateaued, possibly due to the rising rates of smoking and obesity among Chinese youngsters. Comprehensive population-level treatments aimed at smoking cessation, obesity prevention, and gastrointestinal endoscopy screening should be carried out immediately, particularly for men and older birth cohorts at a higher risk of esophageal cancer.

## 1. Introduction

Esophageal cancer is the seventh most frequent cancer in the world and the sixth most common cause of cancer mortality [1]. Histologically, esophageal cancer is split into two main types: esophageal squamous cell carcinoma (ESCC, which accounts for about 84% of cases) and esophageal adenocarcinoma (EAC, which accounts for approximately 15% of cases) [2]. Despite reductions in age-standardized mortality rates (ASMRs) and death coinciding with the introduction of a variety of preventative interventions and treatments, esophageal cancer continues to be a primary cause of cancer mortality and burden worldwide [3]. As a result, analyzing the relationship between long-term trends in esophageal cancer mortality is crucial to evaluating the impact of public health control policies in various regions [4].

It was anticipated that 604,000 new patients with esophageal cancer and 544,000 fatalities would occur worldwide in the year 2020 [1]. The burden of esophageal cancer varies significantly around the globe, with East Asia having the highest age-standardized incidence rates and the second-highest ASMRs [3]. The mortality of esophageal cancer in China and Japan dropped between 1990 and 2017, following global trends. Nonetheless, both countries had an increase in deaths over this time [5]. As located in East Asia, China and Japan possess similar racial backgrounds and parallel cultures. In addition, both countries are facing a heavy burden of esophageal cancer. Nevertheless, Japan has advanced over China in terms of industrialization and urbanization, as well as having more readily available and accessible medical care. Comparing the burden and trends of esophageal cancer between China and Japan could benefit resource planning and allocation, along with providing underlying etiological insights.

The age-period-cohort (APC) model is a widely used statistical tool for depicting and interpreting secular trends in disease mortality across time [6,7]. A new understanding of the esophageal cancer burden can be gained by adopting the APC analysis to disentangle overall mortality trends. As far as we know, research comparing esophageal cancer mortality trajectories between China and Japan using the same APC model has rarely been performed. We extracted data from the Global Burden of Disease Study (GBD) 2019 to examine the longitudinal changes in esophageal cancer mortality in China and Japan from 1990 to 2019. This study enabled us to more accurately compare and evaluate the temporal trends in esophageal cancer and the driving forces underlying these secular trends. The APC analysis was also utilized to look at the independent influences of chronological age, temporal period, and birth cohort.

## 2. Materials and Methods

### 2.1. Data Sources

The Global Burden of Diseases (GBD), which was directed by the Institute for Health Metrics and Evaluation (IHME) (http://ghdx.healthdata.org/gbd-results-tool (accessed on 1 June 2022)), at the University of Washington, is the most thorough systematic epidemiological study in the world to date. GBD creates a unique platform to compare the magnitude of diseases, injuries, and risk factors across age groups, sexes, countries, regions, and time. For decision makers, health sector leaders, researchers, and informed citizens, the GBD approach provides an opportunity to compare their countries’ health progress to that of other countries, and to understand the leading causes of health loss that could potentially be avoided. GBD 2019 offers a comprehensive analysis of 369 injuries and diseases, and 87 risk factors among 204 countries and territories [8,9].

Information on esophageal cancer in China and Japan originated from the GBD 2019. Censuses, the Disease Surveillance Point system, the Maternal and Child Surveillance System, and the Chinese Center for Disease Control and Prevention Cause of Death Reporting System were the critical data sources in China [10]. In comparison, Japan’s Social Health Insurance System was primarily responsible for providing data on esophageal cancer mortality [11]. The Cause of Death Ensemble model (CODEm) was adopted to estimate mortality based on initial data sources and confounding factors, including smoking prevalence, alcohol consumption, education, the socio-demographic index, and so on [9]. The temporal trends of mortality rates for esophageal cancer were age-standardized to the GBD 2019 global standard population. The calculation formula for age-standardized mortality rates (ASMRs) is as follows:$$\mathrm{ASMRs}\;=\frac{\sum \begin{array}{c} \mathrm{Age}\;\mathrm{composition}\;\mathrm{of}\;\mathrm{standard}\;\mathrm{group}\;\mathrm{population}\\ \times \mathrm{Age}\;\mathrm{specific}\;\mathrm{mortality} \end{array}}{\mathrm{Age}\;\mathrm{composition}\;\mathrm{of}\;\mathrm{standard}\;\mathrm{population}}$$

### 2.2. Statistical Analysis

In this research, the APC analysis was employed to investigate the links between the effects of age, period, birth cohort, and esophageal cancer mortality [12]. The APC model is a Poisson-based log-linear regression model in which the dependent variable is mortality, and the independent variables are chronological age, temporal period, and birth cohort. The essential idea behind this model is to ascertain the values of age, period, and cohort effect by fitting the regression connection between the target event’s incidence rate and age, period, and birth cohort. Age effects reflect variations in mortality risk brought on by physiological changes, social experience, and changes in social status as people become older. Period effects represent the diversity in risk associated with a specific period or year that affects all age groups. Cohort effects refer to the various risks of outcome accumulated by individuals in different birth cohorts who experience diverse social and historical events at the same age. The net drifts indicate the annual percentage change after controlling for period and cohort effects. The local drifts represent the annual percentage change in the anticipated age-specific rates over time. The cross-sectional age curves reveal the expected age-specific rates in the reference period adjusted for the cohort effect. The longitudinal age curves reveal the expected age-specific rates in the reference cohort adjusted for period effects. The calculated parameters were acquired by utilizing the National Cancer Institute’s age-period-cohort Web Tool [12]. The parameters are combined to produce functions that describe relationships between the mortality rate of esophageal cancer and attained age, calendar period, and birth cohort.

The basic age-period-cohort model was exhibited as:$$\rho =\log\left(\theta_{apc}/N_{apc}\right)=\alpha_{a}+\pi_{p}+\gamma_{c}$$
where *ρ* represents the log transformation of the expected mortality rate of esophageal cancer. *α*, *π*, and *γ* represent age, period, and cohort effects, respectively.

The longitudinal form and cross-sectional form of the APC model were respectively expressed as:$$\rho_{ac}=\mu +\left(\alpha_{L}+\pi_{L}\right)\left(a-\bar{\;a}\right)+\left(\pi_{L}+\gamma_{L}\right)\left(c-\bar{c}\right)+\tilde{\alpha_{a}}+\tilde{\pi_{p}}+\tilde{\gamma_{c}}$$
$$\rho_{ap}=\mu +\left(\alpha_{L}-\gamma_{L}\right)\left(a-\bar{a}\right)+\left(\pi_{L}+\gamma_{L}\right)\left(p-\bar{p}\right)+\tilde{\alpha_{a}}+\tilde{\pi_{p}}+\tilde{\gamma_{c}}$$
where ($\alpha_{L}+\pi_{L}$) is the longitudinal age trend, ($\alpha_{L}-\gamma_{L}$) is the cross-sectional age trend, ($\pi_{L}+\gamma_{L}$) is the net drift, and $\tilde{\alpha_{a}}$, $\tilde{\pi_{p}}$, and $\tilde{\gamma_{c}}$ are the deviations for age, period, and cohort, respectively.

To estimate the age, period, and cohort impacts, we separated the esophageal cancer age-standardized mortality into five-year age groups ranging from 25–29 years to 80–84 years, with 50–54 years serving as the reference age group. Participants under 25 were omitted from this analysis due to the rarity of esophageal cancer mortality in this age group. The APC framework stipulates that the age and period intervals must be identical. Period data are thus subdivided into five-year segments extending from 1990 to 2019, with the survey years 2000 to 2004 serving as the baseline. Because birth cohort is determined by the subject’s age and the date of occurrence of the event, namely, cohort = period − age, the relevant birth cohorts are 1910–1914 to 1990–1994, using the birth cohorts from 1950 to 1954 as a benchmark. Two-dimensional contingency tables of age-period-specific mortality rates for esophageal cancer in China and Japan were shown in Appendix A. Wald chi-square tests were applied to examine the statistical significance of the assessable parameters and functions. All statistical tests were two-tailed, and statistical significance was defined as *p* < 0.05.

## 3. Results

### 3.1. Trends in Esophageal Cancer Mortality

Figure 1 represents the decreasing trajectory of ASMRs for esophageal cancer mortality by gender and country. Compared with Japan, China has remarkably higher ASMRs per year regardless of gender. Between 1990 and 2019, the ASMRs in China decreased by 28.95% and 59.73% for males and females, respectively. The mortality rates in Japan declined by 22.36% for men and 30.50% for women in the same time frame. However, notwithstanding the overall downward trends, ASMRs in China increased from 1998 to 2004 and then dropped precipitously after 2005. Additionally, esophageal cancer deaths rose by 45.70% in China and 67.22% in Japan from 1990 to 2019 (Table 1). It is interesting to note that the number of deaths among Chinese women hardly changed from 1990 to 2019, whereas its counterpart among Chinese men climbed from 117.08 thousand in 1990 to 197.72 thousand in 2019.

### 3.2. Trends in Age- and Cohort-Specific Esophageal Cancer Mortality

We grouped the mortality and population data into consecutive 5-year periods from 1990 to 1994 (median, 1992) to 2015 to 2019 (median, 2017) and 18 consecutive cohorts, including those born from 1910 to 1914 (median, 1912) to 1990 to 1994 (median, 1992). Figure 2 illustrated age-specific mortality rates of esophageal cancer by period and gender in China and Japan from 1990 to 2019. As shown in Figure 2, the esophageal cancer mortality rate increased with age group and declined between 1990 to 1994 and 2015 to 2019 in both sexes and countries (*p* < 0.01 for all). Cross-sectional age curves present the expected age-specific rates in the reference period, i.e., 2000 to 2004, after adjusting for cohort effects. Figure 3 presented cohort-specific mortality rates of esophageal cancer by age and sex in China and Japan. Among Chinese men and women, esophageal cancer mortality rates first showed an increasing and then a decreased trend over birth cohorts across all age groups. On the contrary, there was a decreasing trend across birth cohorts among Japanese men and women, indicating a relatively lower risk of CVD mortality in cohorts born more recently (*p* < 0.01 for all).

### 3.3. Net Drift and Local Drift in Different Age Groups

Figure 4 and Table 2 illustrate the net drifts and local drifts for esophageal cancer mortality in China and Japan. The net drift was disparate between China and Japan. Between 1990 and 2019, Japanese women witnessed a small decrease in mortality, whilst males encountered a tiny but positive trend (−0.50% [95% CI, −0.91 to −0.08] versus −1.86% [−2.12 to −1.59]). In China, there were significant gender gaps in the overall net drift, with men (−1.98% [−2.22 to −1.74]) experiencing less improvement in mortality than women (−5.22% [−5.77 to −4.68]). Local drifts were less than zero for all age groups regardless of gender in China and Japan, revealing a downwards trend in esophageal cancer mortality over the research period. The curves of local drifts for esophageal cancer mortality across the age groups are shaped like a U-shaped curve for both sexes and countries, except for Japanese women. The most impressive improvement was in the 40–44 age groups in China, −6.65%/year for women and −3.21%/year for men, whereas for Japanese men, the best improvement was in the 45–49 age groups (−3.40%/year).

### 3.4. The Age, Period, and Cohort Effects on Esophageal Cancer Mortality

The APC analysis’s corresponding coefficients, which were reported in Appendix A, were used to estimate the RRs for a specific age, period, or birth cohort, which are shown in Table 3. The longitudinal age curves of esophageal cancer mortality by sex and country are shown in Figure 5. After adjusting for period and cohort effects, we discovered that esophageal cancer mortality rose advancing age, making a peak at the 80–84 age groups. Men in China underwent the highest growth with age, escalating from 0.41 to 200.88 per 100,000; while Chinese women and Japanese men showed a relatively moderate increase in esophageal cancer mortality, climbing from 0.81 to 47.11 per 100,000 and from 0.05 to 65.02 per 100,000 respectively; the raising with age was not apparent for Japanese women, rising from 0.02 to 11.36 per 100,000. Men are more likely to die than women of all ages from esophageal cancer in the two countries. With age, the disparity in death rates between men and women widens.

As can be seen from Figure 6, China and Japan had similar period effects on the esophageal cancer mortality rate for men and women. Both countries’ period effects generally decreased, with China’s reductions being more pronounced than Japan’s over time. We also observed that in China, women experienced a steeper decline than men, whereas in Japan, the reverse is true: males underwent a faster drop than women. The estimated cohort effects on the esophageal cancer mortality rate by gender and country are shown in Figure 7. Likewise, the cohort effects have also positively affected the mortality rate of esophageal cancer for both sexes and countries, with more noticeable decreases for China than for Japan. Cohort effects for men in China fell from 1.43 in 1910 to 0.40 in 1990 and dropped from 3.53 to 0.08 for women, whereas cohort effects for Japanese males and females shrank from 1.36 to 0.41 and from 1.84 to 0.81, respectively. Nonetheless, the improvement of cohort effects for China’s younger generation has slowed or even stagnated. Furthermore, the mortality of esophageal cancer exhibited a statistically significant difference in the period and cohort rate ratio for sexes in China and Japan (*p* < 0.05), as did the net and local drifts, according to the Wald chi-square tests (Appendix A).

## 4. Discussion

From 1990 to 2019, the ASMRs of esophageal cancer reduced in China and Japan, especially in China after 2005. In both countries, esophageal cancer mortality increased considerably from 25 to 84 years for both sexes combined. According to our research, both period and cohort effects dropped steadily in China and Japan. China’s reduction in cohort effects was more visible than Japan’s, but the decline in China’s younger generation has lately leveled off. Furthermore, the findings of this study implied that mortality rates varied by gender.

Although esophageal cancer ASMRs dropped in both China and Japan over the 1990–2019 period, China’s decreases for both sexes are more striking, particularly after 2005. This remarkable drop could be linked to the Health’s Early Detection and Treatment Program of the Chinese Ministry launched in 2005. The program’s first goal was to detect esophageal cancer. Early esophageal cancer generally has no classic symptoms, leading to the majority of patients being diagnosed in the advanced stages [13]. At this time, more than 50 percent of patients suffer nonreversible lesions, resulting in a horrible five-year survival rate of less than 20% [14,15]. However, screening for esophageal cancer can detect precancerous lesions and early malignancies, which can now be treated with more minor invasive procedures such as endoscopic submucosal dissection, endoscopic mucosal resection, neoadjuvant radiation, and chemotherapy [16,17]. According to a study conducted in Cixian County [18], mass upper gastrointestinal endoscopy boosted the survival rate of esophageal cancer patients considerably. Meanwhile, stomach cancer can be detected with mass upper gastrointestinal endoscopy [19]. Hence, we should promote widespread upper gastrointestinal endoscopy in both countries, where esophageal and gastric cancer rates are high. Despite esophageal cancer ASMRs having dropped in China and Japan, esophageal cancer-related fatalities have climbed over the last three decades. This discovery, which corroborated findings from earlier research [3,5], might be brought on by aging and population expansion. It is well known that the aging population of China and Japan has accelerated over the last three decades [20,21]. Additionally, older people have a higher risk of dying from esophageal cancer [22]. As a result of changes in the population’s age and structure in both countries, these trends demonstrate that esophageal cancer still generates a significant disease burden and would impose a higher strain on public health care systems.

Stark differences were detected in the ASMRs of esophageal cancer between China and Japan. This discrepancy is likely due to the heterogeneity in risk prevalence produced by various phases of socioeconomic development. China’s socioeconomic level is lower than Japan’s, and China’s statewide cancer control measures were also implemented later [23,24]. Therefore, Japan has a higher level of general health awareness, greater nutritional levels, and better esophageal cancer detection, diagnosis, and treatment. Albeit China has a heavier esophageal cancer burden than Japan, the improvement of ASMRs for esophageal cancer in China has been more significant than in Japan from 1990 to 2019. Since the 1960s and 1970s, the Chinese government has conducted a good deal of epidemiological research to identify the major risk factors for esophageal cancer. Preventive measures thereby have been adopted to combat the mortality of esophageal cancer [25]. In addition, China’s central government announced the Four Trillion RMB Investment Plan to improve health care in 2008 [26].

The mortality for esophageal cancer in China and Japan rose with age advancing, peaking in the 80–84 age group, suggesting that the burden of esophageal cancer was mainly driven by aging transition [27,28]. Previous studies have indicated that the reasons why older people have higher mortality of esophageal cancer are probably due to more prolonged exposure to risk factors and a reduction in physiological function [29]. All these factors may contribute to the uptrend in mortality of esophageal cancer in both countries, as well as South Korea and Brazil, where rising age trends in mortality rates were observed [5,30]. Our results also elucidated that a strong male predominance is a characteristic of the mortality of esophageal cancer, which is consistent with earlier studies [1,31]. For example, the number of deaths from esophageal cancer is considerably higher in men than in women, particularly in China, where men were responsible for nearly all of the rise in esophageal cancer fatalities over the course of the study. The precise reasons for the essentially constant number of deaths among Chinese women are difficult to ascertain. It is possible that the negative impacts of the aging population are counterbalanced by preventive factors for esophageal cancer mortality, such as improved esophageal cancer diagnosis and treatment, higher health awareness, and decreased female smoking rates [17,32,33]. The primary risk factors for esophageal cancer are smoking and heavy drinking [34,35]. It is well known that these bad habits have been more prevalent in men than in women. This might, at least to some extent, explain the higher mortality and deaths of esophageal cancer in men than in women. Thus, the management of aging and risk factors may still be instant measures to be improved for controlling esophageal cancer mortality.

Albeit the period and cohort effects of esophageal cancer have been on a descending trend, it is not easy to give an integrated explanation separately. Obviously, the period effect has generally decreased in China and Japan over the 1990–2019 period. These declines we found might be from the economic gains and dietary improvements. Current research revealed a negative correlation between socioeconomic status and ESCC [36,37,38], the most predominant histological type in China and Japan [39]. Low socioeconomic levels are on behalf of multiple correlated and interrelated variables such as drinking unpiped water and high indoor air pollution. Indeed, previous studies have indicated that exposure to indoor air pollution and unavailable piped water is linked to an elevated risk of ESCC [40]. Numerous studies have shown that exposure to indoor air pollution over an extended period of time can raise the risk of esophageal cancer [41,42,43]. Abnet et al. found that polycyclic aromatic hydrocarbons (from indoor air pollution) may be a pathogenic factor of esophageal cancer [44]. Polycyclic aromatic hydrocarbons, which could be taken in by the body and circulated in the bloodstream, can boost DNA methylation and epigenetic alterations. What is intriguing about this article is that there is a quicker declination of period effect for Chinese women than for men, while the decrease for Japanese men is more favorable than for women. This apparent discordance between China and Japan may be associated with the difference in improvements in smoking. The age-standardized smoking prevalence fell significantly for both genders and countries between 1990 and 2015. More specifically, the reduction in smoking prevalence for males is worse than for females in China (−22.4% versus −48.4 %). In contrast, Japanese men’s decline in smoking prevalence is better than women’s (−44.8% versus −16.0%) [32]. This gap in the smoking rate reduction can also be utilized to interpret the gender disparity in net drift between the two nations. In other words, net drift in Chinese men is lower than in women, whereas net drift in Japanese men is higher than in women. Moreover, Japanese males’ drinking rates fell from 54.9% in 1995 to 31.6% in 2018, while Japanese females’ drinking rates grew from 7.8% in 1995 to 10.4% in 2018 [45]. This may also be the explanation why Japanese men have shown greater improvement in esophageal cancer mortality than Japanese women.

The cohort effects on esophageal cancer mortality in China and Japan revealed a generally declining trajectory from the 1910–1914 to 1990–1994 birth cohorts. Because of improved education and lifestyle, younger generations may have a greater consciousness of health and illness prevention than older generations [33]. Obviously, China has a more remarkable decline in cohort effects than Japan; regrettably, the reduction in cohort effects for China’s younger generation has slowed or stalled. Albeit the underlying mechanisms of this stagnation are unclear, the following are some plausible explanations. On the one hand, the smoking rates of male juveniles increased from 16.0% to 23.5% between 2003 and 2013, while female adolescent smoking rates went from 0.4% to 1.1% in China [46]. On the other hand, obesity in children and young adults has been more common in China over the past thirty years [47], which may contribute to an exacerbation in EAC mortality, as obesity is a key risk factor for EAC [48]. Considering that there is no predominant risk factor for esophageal cancer, comprehensive measures should be taken to reduce the risk factors mentioned above, especially since most of these risk factors are also severe risk factors (such as smoking and obesity) for numerous typical chronic diseases.

This study had several limitations. First, ESCC and EAC, two major histological subtypes of esophageal cancer, have distinguishing risk factors and mortality tendencies [49]. Second, the mortality of esophageal cancer varies significantly within a country, and there are sharp discordances between rural and urban areas, especially in China [3,39,44]. However, China and Japan only have country-level data, not more granular geographic unit data in GBD 2019. Finally, like other APC research, analyses at the group level inherently entail an ecological fallacy. Herein, the conclusions obtained from this research must be validated through individual studies.

## 5. Conclusions

In summary, this article firstly compared the long-term trends in esophageal cancer mortality in China and Japan and evaluated the independent impacts of age, period, and cohort effects from 1990 to 2019. We elucidated that the ASMRs, period RR, and cohort RR have all declined for both genders and countries over the last thirty years. Overall, men and the elderly were the most at risk for esophageal cancer mortality. Even though China’s disease burden of esophageal cancer is substantially higher than Japan’s, China achieved a more impressive reduction in esophageal cancer mortality. However, for China’s younger generation, the lowering of cohort effects has slowed or even stopped, probably due to the rising rates of smoking and obesity among Chinese youths. Meanwhile, smoking remained a major public health concern in China and Japan. Systematic actions should be addressed to reduce these known and potential risk factors. It would also be recommended to lower the gastrointestinal endoscopy screening age to discover esophageal cancer precancerosis as early as possible.

## Figures and Tables

**Figure 1 ijerph-19-10302-f001:**
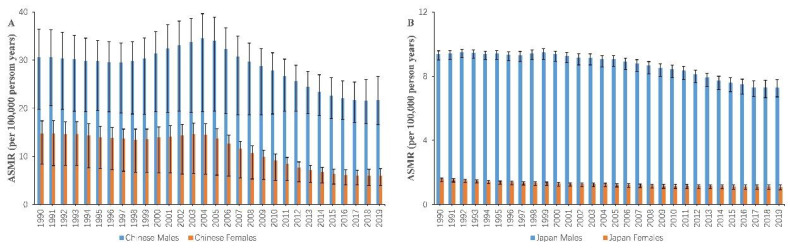
Trends in the ASMRs of esophageal cancer mortality by gender in China (**A**) and Japan (**B**) from 1990 to 2019, at all ages. ASMR: age-standardized mortality rate. Error bars represented the 95% CIs for esophageal cancer mortality.

**Figure 2 ijerph-19-10302-f002:**
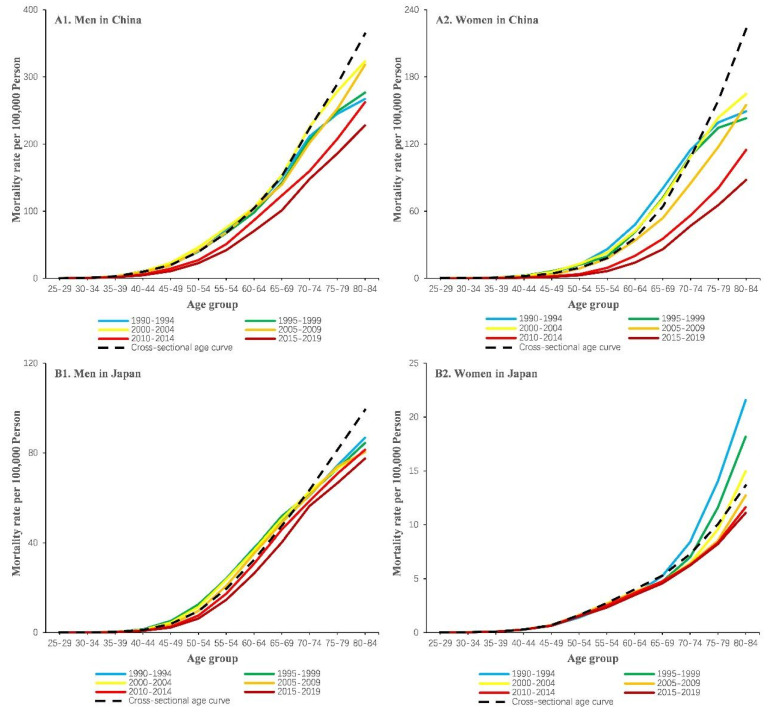
Age-specific mortality rates of esophageal cancer by period and sex in China (**A1**,**A2**) and Japan (**B1**,**B2**), 1990 to 2019. The study period was organized into 5-year periods, namely, 1990–1994 (median, 1992), 1995–1999 (median, 1995), 2000–2004 (median, 2002), 2005–2009 (median, 2007), 2010–2014 (median, 2012), and 2015–2019 (median, 2017). Cross-sectional age curves present the expected age-specific rates in the reference period, i.e., 2000 to 2004, after adjusting for cohort effects (*p* < 0.01 for all).

**Figure 3 ijerph-19-10302-f003:**
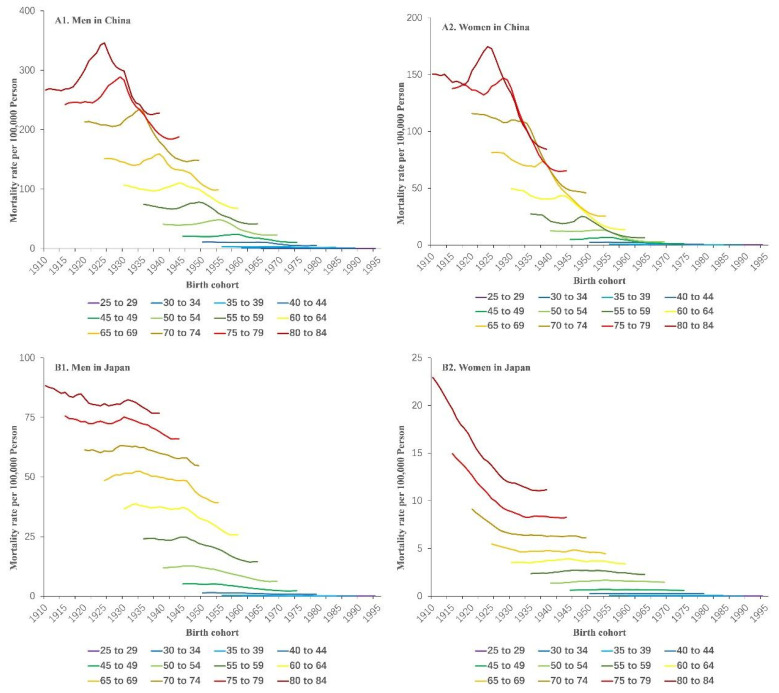
Cohort-specific mortality rates of esophageal cancer by age and sex in China (**A1**,**A2**) and Japan (**B1**,**B2**), 1990 to 2019. Data on esophageal cancer mortality were arranged into 18 successive birth cohorts, which include those born between 1910–1914 (median, 1912) and 1990–1994 (median, 1992), and 12 age groups between 20–24 (median, 22 years) and 80–84 years of age (*p* < 0.01 for all).

**Figure 4 ijerph-19-10302-f004:**
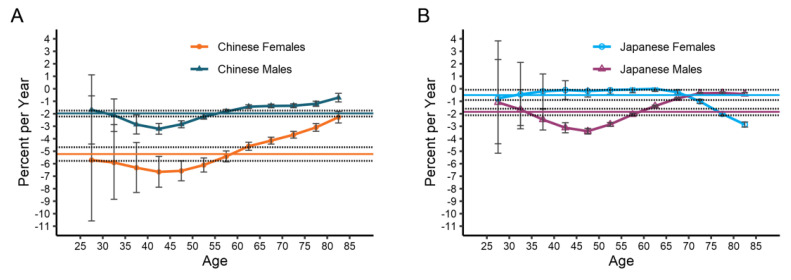
The local drift with net drift values for esophageal cancer mortality for males and females in China (**A**) and Japan (**B**). Net drift values are depicted as solid lines with dashed lines representing their 95% CI. Error bars represent the 95% CI for the local drift values.

**Figure 5 ijerph-19-10302-f005:**
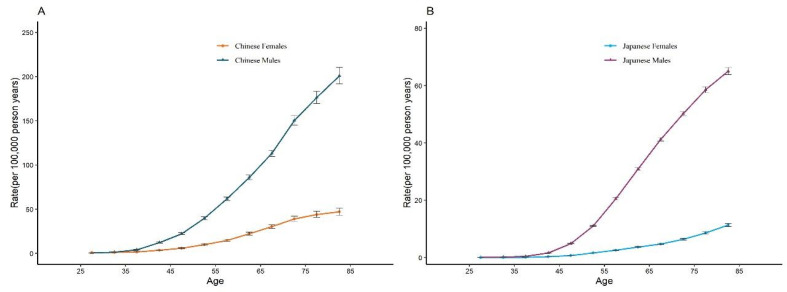
Longitudinal age curves of esophageal cancer mortality by genders in China (**A**) and Japan (**B**). Fitted longitudinal age-specific rates of esophageal cancer mortality rates (per 100,000 person-years) and the corresponding 95% CI.

**Figure 6 ijerph-19-10302-f006:**
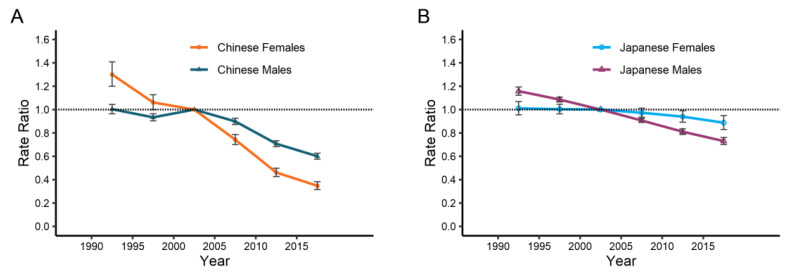
Period effects of esophageal cancer mortality by genders in China (**A**) and Japan (**B**). The relative risk of each period compared with the reference period (2000–2004) adjusted for age and nonlinear cohort effects and the corresponding 95% CI.

**Figure 7 ijerph-19-10302-f007:**
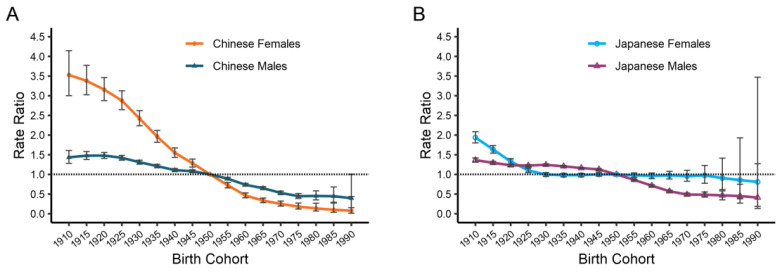
Cohort effects of esophageal cancer mortality by genders in China (**A**) and Japan (**B**). The relative risk of each cohort compared with the reference cohort (the 1950s) adjusted for age and nonlinear period effects and the corresponding 95% CI.

**Table 1 ijerph-19-10302-t001:** Esophageal Cancer mortality in China and Japan between 1990 and 2019.

	China		Japan	
	1990	2019	1990	2019
Both				
ASMR ^a^, per 100,000	22.08263	13.14883	4.931302	3.911883
Deaths, ×1000	176.6016	257.3158	8.37199	13.99966
Relative proportion in all causes of death, %	2.11%	2.41%	1.03%	1.00%
Men				
ASMR, per 100,000	30.52923	21.6905	9.356257	7.264125
Deaths, ×1000	117.0849	197.7159	6.861007	11.55804
Relative proportion in all causes of death, %	2.55%	3.17%	1.56%	1.62%
Women				
ASMR, per 100,000	14.69056	5.916205	1.570562	1.091518
Deaths, ×1000	59.51674	59.59982	1.510983	2.441627
Relative proportion in all causes of death, %	1.58%	1.35%	0.41%	0.35%

Note: ^a^ ASMR: age-standardized mortality rate.

**Table 2 ijerph-19-10302-t002:** The local drifts and net drifts for China and Japan (%).

Drift	China		Japan	
	Men	Women	Men	Women
Local Drift ^a^ (95%CI)				
Age				
25–29	−1.69 (−4.42 to 1.12)	−5.71 (−10.58 to −0.57)	−1.09 (−4.40 to 2.34)	−0.77 (−5.16 to 3.83)
30–34	−2.13 (−3.42 to −0.81)	−5.91 (−8.85 to −2.87)	−1.66 (−3.20 to −0.08)	−0.45 (−2.94 to 2.11)
35−39	−2.86 (−3.62 to −2.10)	−6.32 (−8.30 to −4.30)	−2.47 (−3.30 to −1.64)	−0.21 (−1.58 to 1.19)
40–44	−3.21 (−3.64 to −2.77)	−6.65 (−7.88 to −5.41)	−3.12 (−3.53 to −2.71)	−0.10 (−0.85 to 0.64)
45−49	−2.84 (−3.12 to −2.56)	−6.57 (−7.37 to −5.77)	−3.40 (−3.62 to −3.17)	−0.18 (−0.66 to 0.30)
50−54	−2.23 (−2.43 to −2.02)	−6.11 (−6.67 to −5.54)	−2.85 (−3.00 to −2.71)	−0.10 (−0.43 to 0.24)
55−59	−1.83 (−2.00 to −1.65)	−5.42 (−5.85 to −4.98)	−2.06 (−2.16 to −1.96)	−0.06 (−0.32 to 0.20)
60−64	−1.43 (−1.59 to −1.28)	−4.60 (−4.93 to −4.28)	−1.38 (−1.46 to −1.31)	0.00 (−0.21 to 0.21)
65–69	−1.38 (−1.52 to −1.23)	−4.15 (−4.43 to −3.88)	−0.77 (−0.83 to −0.71)	−0.27 (−0.45 to −0.09)
70–74	−1.36 (−1.52 to −1.20)	−3.68 (−3.95 to −3.41)	−0.36 (−0.43 to −0.30)	−1.01 (−1.17 to −0.84)
75–79	−1.20 (−1.41 to −1.00)	−3.09 (−3.40 to −2.78)	−0.34 (−0.42 to −0.26)	−2.01 (−2.18 to −1.85)
80–84	−0.71 (−1.05 to −0.36)	−2.28 (−2.74 to −1.81)	−0.42 (−0.54 to −0.30)	−2.86 (−3.07 to −2.66)
Net Drift ^a^ (95%CI)				
	−1.98 (−2.22 to −1.74)	−5.22 (−5.77 to −4.68)	−1.86 (−2.12 to −1.59)	−0.50 (−0.91 to −0.08)

Note: ^a^ CI confidence interval; Net drifts represent the overall annual percentage change in the age-standardized rate based on period and birth cohort. Local drifts indicate the annual percentage change over time specific to the age group.

**Table 3 ijerph-19-10302-t003:** The relative risks due to age, period, and cohort effects in China and Japan.

Factor	China				Japan			
	Men		Women		Men		Women	
	RR ^a^	95%CI ^b^	RR	95%CI	RR	95%CI	RR	95%CI
Age								
25–29	0.41	0.29 to 0.58	0.81	0.48 to 1.35	0.05	0.04 to 0.08	0.02	0.01 to 0.03
30–34	1.44	1.20 to 1.73	1.21	0.85 to 1.74	0.16	0.13 to 0.20	0.03	0.02 to 0.04
35–39	4.05	3.65 to 4.50	1.68	1.29 to 2.20	0.41	0.36 to 0.46	0.09	0.07 to 0.11
40–44	12.17	11.44 to 12.95	3.51	2.97 to 4.14	1.60	1.51 to 1.69	0.29	0.26 to 0.32
45–49	22.31	21.28 to 23.40	5.95	5.28 to 6.70	4.86	4.71 to 5.01	0.68	0.63 to 0.73
50–54	39.67	38.22 to 41.18	9.97	9.11 to 10.91	10.99	10.77 to 11.22	1.57	1.48 to 1.66
55–59	61.68	59.69 to 63.73	14.70	13.61 to 15.89	20.49	20.15 to 20.82	2.58	2.46 to 2.70
60–64	85.95	83.32 to 88.66	22.24	20.70 to 23.90	30.87	30.42 to 31.33	3.69	3.54 to 3.85
65–69	113.14	109.62 to 116.77	30.34	28.27 to 32.57	41.21	40.62 to 41.80	4.74	4.55 to 4.94
70–74	150.47	145.06 to 156.08	39.08	36.16 to 42.25	50.19	49.37 to 51.02	6.38	6.09 to 6.69
75–79	176.38	169.50 to 183.54	43.99	40.56 to 47.71	58.53	57.52 to 59.56	8.57	8.17 to 8.99
80–84	200.88	191.80 to 210.39	47.11	43.17 to 51.41	65.02	63.79 to 66.28	11.36	10.81 to 11.93
Period								
1990–1994	1.00	0.96 to 1.04	1.30	1.20 to 1.41	1.16	1.12 to 1.19	1.01	0.95 to 1.07
1995–1999	0.93	0.90 to 0.96	1.06	1.00 to 1.13	1.09	1.07 to 1.11	1.00	0.96 to 1.05
2000–2004	1.00	1.00 to 1.00	1.00	1.00 to 1.00	1.00	1.00 to 1.00	1.00	1.00 to 1.00
2005–2009	0.90	0.87 to 0.93	0.74	0.70 to 0.79	0.91	0.89 to 0.92	0.97	0.94 to 1.01
2010–2014	0.71	0.68 to 0.73	0.46	0.43 to 0.50	0.81	0.79 to 0.83	0.94	0.89 to 0.99
2015–2019	0.60	0.58 to 0.63	0.35	0.32 to 0.38	0.73	0.70 to 0.76	0.89	0.83 to 0.95
Cohort								
1910–1914	1.43	1.28 to 1.61	3.53	3.00 to 4.15	1.36	1.31 to 1.42	1.94	1.80 to 2.09
1915–1919	1.48	1.38 to 1.58	3.38	3.02 to 3.77	1.29	1.26 to 1.33	1.64	1.54 to 1.74
1920–1924	1.48	1.41 to 1.56	3.16	2.88 to 3.46	1.23	1.21 to 1.26	1.33	1.26 to 1.40
1925–1930	1.42	1.36 to 1.48	2.88	2.65 to 3.13	1.23	1.20 to 1.25	1.10	1.05 to 1.16
1931–1934	1.31	1.26 to 1.36	2.42	2.24 to 2.62	1.25	1.22 to 1.27	1.00	0.95 to 1.05
1935–1939	1.21	1.17 to 1.26	1.96	1.82 to 2.12	1.21	1.19 to 1.23	0.98	0.94 to 1.03
1940–1944	1.11	1.07 to 1.15	1.55	1.43 to 1.67	1.16	1.14 to 1.18	0.98	0.94 to 1.03
1945–1949	1.08	1.04 to 1.12	1.28	1.19 to 1.39	1.13	1.11 to 1.15	1.00	0.96 to 1.05
1950–1954	1.00	1.00 to 1.00	1.00	1.00 to 1.00	1.00	1.00 to 1.00	1.00	1.00 to 1.00
1955–1959	0.89	0.86 to 0.93	0.72	0.66 to 0.80	0.86	0.84 to 0.88	0.98	0.92 to 1.04
1960–1964	0.74	0.70 to 0.77	0.46	0.40 to 0.53	0.71	0.69 to 0.74	0.96	0.89 to 1.04
1965–1969	0.65	0.61 to 0.69	0.34	0.28 to 0.40	0.57	0.55 to 0.60	0.98	0.88 to 1.08
1970–1974	0.53	0.49 to 0.58	0.25	0.19 to 0.32	0.49	0.45 to 0.53	0.96	0.83 to 1.11
1975–1979	0.45	0.39 to 0.51	0.18	0.12 to 0.27	0.48	0.42 to 0.55	0.98	0.77 to 1.23
1980–1984	0.45	0.35 to 0.58	0.14	0.07 to 0.27	0.47	0.35 to 0.62	0.91	0.58 to 1.41
1985–1989	0.45	0.29 to 0.68	0.10	0.04 to 0.27	0.45	0.27 to 0.75	0.86	0.38 to 1.93
1990–1994	0.40	0.16 to 1.00	0.08	0.01 to 0.44	0.41	0.13 to 1.27	0.81	0.19 to 3.47

Note: ^a^ RR is the abbreviation of the rate ratio; ^b^ 95% CI is the abbreviation of 95% confidence interval.

## Data Availability

Data presented in this study are openly available from the Global Burden of Disease. Available online: http://ghdx.healthdata.org/gbd-results-tool (accessed on 1 June 2022).

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
