# Peer review of "Comparison of Secular Trends in Esophageal Cancer Mortality in China and Japan during 1990–2019: An Age-Period-Cohort Analysis"

_ijerph, 2022, doi:10.3390/ijerph191610302_

Round 1

Reviewer 1 Report

The work is interesting, the text is clear, the discussion sounds good.

The data in table 1, with the reduction of the ASMRs, but the increase in deaths, deserves to be discussed more.

There is an inconsistency:

Pag 1, line 40: ASMRs= age-standardized morbidity rates

Pag 2, line 92: ASMRs= age-standardized mortality rates

Reviewer 2 Report

The present study aims to compare esophageal cancer (EC) mortality in China and Japan during the last thirty years. It is very interesting study since esophageal cancer has a high mortality rate and any information on this type of cancer could support public health systems. It should also be noted that esophageal cancer produces no symptoms at early stages and therefore such studies are needed to introduce new directions for its prevention.

The text is well-written. There are some minor typo/grammar errors (i.e., line 138: “Tables” instead of “Table”) that should be corrected. In addition, there are several points requiring correction and/or clarification, as follows.

11.       Line 37: use one of the two refs.

22.       Heading 3.1 and Table 1: According to Table 1, almost the same number of women died of EC in 1990 (59.5) and 2019 (59.6). Therefore, all increase in deaths is due to men. The authors should check their findings. If this is true, it should be presented and discussed in text. Moreover, the difference in deaths’ increase between China and Japan is due exclusively to the limited increase of women deaths and this should also be discussed.

33.  .    Figure 5: The corresponding CI is shown only for China. In addition, the raising with age (from 0.02 to 11.36 per 100,000) will be apparent after changing the range of y-axis.

44.       Lines 197-198: Check your findings. If all are correct, provide an explanation whether in China women experienced a steeper decline than men, whereas in Japan males underwent a faster drop than women.

55.       Line 282: 1900-1904 birth cohort should be 1910-1914.

66.       Line 313: The authors should consider air pollution as an additional factor for EC.
